# MYCN and Metabolic Reprogramming in Neuroblastoma

**DOI:** 10.3390/cancers14174113

**Published:** 2022-08-25

**Authors:** Mohit Bansal, Anamika Gupta, Han-Fei Ding

**Affiliations:** Division of Molecular and Cellular Pathology, Department of Pathology, O’Neal Comprehensive Cancer Center, Heersink School of Medicine, University of Alabama at Birmingham, Birmingham, AL 35294, USA

**Keywords:** cancer metabolism, metabolic reprogramming, MYCN, neuroblastoma, pediatric cancer

## Abstract

**Simple Summary:**

Metabolic reprogramming has a central role in the initiation and progression of cancer, including high-risk neuroblastoma, a deadly childhood malignant tumor of the sympathetic nervous system. This rewiring of cellular metabolism increases the manufacture of fuel and building blocks for biomass production, which is essential to sustain the growth and proliferation of neuroblastoma cells. However, the rewiring also makes neuroblastoma cells metabolically distinct from normal sympathetic neurons, thereby offering new therapeutic opportunities. In this review, we summarize the recent progress in the study of neuroblastoma metabolic reprogramming and underlying molecular mechanisms.

**Abstract:**

Neuroblastoma is a pediatric cancer responsible for approximately 15% of all childhood cancer deaths. Aberrant MYCN activation, as a result of genomic *MYCN* amplification, is a major driver of high-risk neuroblastoma, which has an overall survival rate of less than 50%, despite the best treatments currently available. Metabolic reprogramming is an integral part of the growth-promoting program driven by MYCN, which fuels cell growth and proliferation by increasing the uptake and catabolism of nutrients, biosynthesis of macromolecules, and production of energy. This reprogramming process also generates metabolic vulnerabilities that can be exploited for therapy. In this review, we present our current understanding of metabolic reprogramming in neuroblastoma, focusing on transcriptional regulation as a key mechanism in driving the reprogramming process. We also highlight some important areas that need to be explored for the successful development of metabolism-based therapy against high-risk neuroblastoma.

## 1. Introduction

Cellular metabolism provides the bioenergetic and biosynthetic needs of living cells [1,2,3]. In quiescent cells, such as differentiated neurons, a major function in the cellular metabolism is to convert nutrients (e.g., glucose) into energy (e.g., ATP) to sustain cellular processes. This is accomplished primarily via glycolysis and the tricarboxylic acid (TCA) cycle in combination with electron transport and oxidative phosphorylation (OXPHOS). Glycolysis converts glucose to pyruvate, generating two net ATP molecules per molecule of glucose. Pyruvate is then oxidized to acetyl coenzyme A (acetyl-CoA), which enters the TCA cycle to generate NADH and FADH_2_. These high-energy electron carriers transfer their electrons to the electron transport chain to support OXPHOS, generating up to 34 more ATP molecules per molecule of glucose. By contrast, proliferating cells also need to synthesize nucleic acids, proteins, and lipids to support growth and generation of daughter cells. This requires reprogramming of cellular metabolism to increase the uptake and catabolism of nutrients to fuel macromolecule biosynthesis (Figure 1). For example, increased glucose uptake and glycolysis in combination with upregulation of glucose-6-phosphate dehydrogenase (G6PD) promote flux of the glycolytic intermediate glucose-6-phosphate through the pentose phosphate pathway to increase the production of ribose-5-phosphate for nucleotide synthesis. Upregulation of phosphoglycerate dehydrogenase (PHGDH) diverts the glycolytic intermediate 3-phosphoglycerate to the serine, glycine, one-carbon metabolic network to produce serine, glycine, and one-carbon units that are required for producing proteins, nucleotides, and lipids. The TCA cycle is also a major source of biosynthetic intermediates for producing amino acids and lipids. Increased glutamine uptake and catabolism provide a steady supply of nitrogen to produce nucleotides and amino acids and of α-ketoglutarate to drive the TCA cycle. The TCA intermediate oxaloacetate can be used to produce aspartate and asparagine. Further, citrate from the TCA cycle can be exported to the cytosol and cleaved by ATP citrate lyase (ACLY) to produce cytosolic acetyl-CoA for the synthesis of fatty acids and cholesterol (Figure 1). 

In this review, we discuss the major metabolic pathways that are activated in neuroblastoma via transcriptional regulation, focusing on MYCN and pathway-specific transcription regulators in this metabolic rewiring process.

## 2. MYCN and Neuroblastoma

Neuroblastoma accounts for approximately 7% of all childhood cancers and 15% of all childhood cancer deaths. It is a malignant tumor of the sympathetic nervous system, arising primarily in abdominal sympathetic ganglia and adrenal medulla, a specialized sympathetic ganglion [4,5,6,7]. Both tissues are derived from neural crest cells, a transient, highly migratory population of multipotent stem cells or progenitor cells [5,8,9]. Neuroblastoma patients are classified into low-, intermediate-, and high-risk groups based upon the age at diagnosis, tumor histopathology and differentiation, DNA index (ploidy), 11q deletion, and *MYCN* amplification status. High-risk patients have an overall survival rate below 50%, despite best current treatments [10,11,12]. Identification of metabolic vulnerabilities of high-risk neuroblastoma may suggest new therapeutic targets and strategies.

Genomic amplification of the oncogene *MYCN* is a key oncogenic event in the development of high-risk neuroblastoma, which occurs in approximately half of all high-risk cases. *MYCN* was originally identified as an amplified gene, homologous to the myelocytomatosis viral oncogene *v-myc*, in neuroblastoma cell lines [13,14]. It was soon discovered that *MYCN* amplification predicts poor prognosis in neuroblastoma patients [15,16]. *MYCN* amplification is most often present at diagnosis and is rarely acquired during progression [4,17], suggesting that *MYCN* amplification is an early and initiating event driving the development of high-risk tumors. This view is supported by laboratory evidence. *TH-MYCN* transgenic mice express human *MYCN* in migrating neural crest cells or sympathetic progenitors under the control of the rat tyrosine hydroxylase (TH) gene promoter. These mice develop neuroblastoma tumors that molecularly and histologically resemble human tumors with *MYCN* amplification [18,19,20]. In *TH-MYCN* mice, neuroblastoma development begins with multifocal hyperplasia in sympathetic ganglia during the first weeks after birth [21]. The hyperplastic lesions are composed predominantly of highly proliferative neuronal progenitors expressing Phox2B (paired like homeobox 2b) [22], a master lineage transcription factor that is expressed in sympathetic progenitors and is essential for embryonic development of the sympathetic nervous system [23]. Additional studies provide further evidence that elevated expression of MYCN is sufficient to drive high-risk neuroblastoma in various animal models [24,25,26]. In addition to neuroblastoma, increased MYCN activation, mainly through genomic amplification, is commonly observed in neuronal and neuroendocrine cancers [17,27], including medulloblastoma [28,29], Wilms tumor [30], retinoblastoma [31], neuroendocrine prostate cancer [32,33,34], glioblastoma [35,36], and small-cell lung cancer [37,38,39].

MYCN is a member of the MYC family of oncogenic transcription factors that also include MYC and MYCL [17,40,41]. Knockout of either *MYC* or *MYCN* in mice results in embryonic lethality [42,43,44], demonstrating that MYCN and MYC have distinct physiological functions, most likely because of their distinct expression patterns during development [45,46]. Nevertheless, MYCN can functionally replace MYC in murine development [47], indicating that the two proteins are, to a large degree, functionally redundant, sharing similar biochemical and biological activities. Like other members of the MYC family, a primary function of MYCN is to promote cell growth and proliferation [48]. In cultured cells, MYCN overexpression induces DNA replication and increases the rate of cell proliferation [49,50]. Conversely, MYCN knockdown triggers cell cycle arrest in *MYCN*-amplified neuroblastoma cell lines [51]. There is also evidence for a physiological function of MYCN in promoting cell proliferation during development. Targeted deletion of MYCN in mouse neural stem and progenitor cells reduces brain size and the proliferation of cerebellar granule neural precursors, with a striking decrease in S-phase and mitotic cells [52,53]. Reprogramming of cellular metabolism is a key mechanism by which MYCN sustains its cell growth and proliferation program.

MYCN and other members of the MYC family are basic-helix-loop-helix-leucine zipper (bHLH-LZ) proteins that can form a heterodimer with other bHLH-LZ proteins, including MYC-associated X (MAX) and MYC-interacting zinc-finger protein 1 (MIZ1, also known as ZBTB17, zinc finger and BTB domain containing 17). In general, the MYC-MAX heterodimer activates and MYC-MIZ1 represses the transcription of their target genes [40,41,48,54]. MYC or MYCN-containing heterodimers bind genomic DNA at the consensus enhancer-box (E-box) sequence CANNTG [48,55] and regulate transcription of a large number of genes that encode proteins involved in fundamental cellular processes, including proliferation, growth, ribosome biogenesis, and metabolism [17,40,41,56]. In addition to the classic model of MYCN-mediated transcriptional regulation, recent evidence indicates that *MYCN* amplification or high-level expression can drive global transcriptional amplification to increase the production of transcripts from all active genes in neuroblastoma cells, which is mediated by MYCN invading the promoters and enhancers of active genes [57]. Since many enzyme-encoding genes are actively transcribed in cells for the maintenance of basic metabolic functions, it is reasonable to speculate that the MYCN-driven global transcriptional amplification could have a profound impact on metabolic reprogramming in neuroblastoma cells.

## 3. Metabolic Reprogramming in Neuroblastoma

### 3.1. Aerobic Glycolysis in Neuroblastoma

Aerobic glycolysis, also known as the Warburg effect, describes the metabolic phenotype of increased glucose uptake and lactate production in the presence of abundant oxygen [58,59,60,61] (Figure 1). It has long been recognized that neuroblastoma tumors display increased glucose uptake, as determined by ^18^F-fluorodeoxyglucose positron emission tomography (FDG-PET) imaging [62]. In addition, it has been shown that increased glycolysis promotes the survival and growth of neuroblastoma cells in vitro and in vivo [63]. Both MYCN and hypoxia-inducible factor 1α (HIF-1α) are required for sustaining the aerobic glycolysis phenotype in neuroblastoma cells by transcriptional upregulation of glycolytic enzymes, including hexokinase 2 (HK2) and lactate dehydrogenase A (LDHA) [64]. More recent studies further confirm a critical role for MYCN in promoting glycolysis in neuroblastoma cells [65,66]. These findings are consistent with the extensive evidence for MYC as a key transcriptional activator of genes involved in glycolysis [67].

Interestingly, recent studies from the Tong laboratory revealed novel mechanisms underlying the increased glycolysis in neuroblastoma cells [68,69]. Through a combination of bioinformatics and overexpression and knockdown studies, Li et al. identified CUT-like homeobox 1 (CUX1) as a transcription factor that upregulates glycolytic genes to promote aerobic glycolysis in neuroblastoma cells. Moreover, they showed that *circ-CUX1*, a circular RNA generated from *CUX1* exon 2 and part of intron 2, can interact with EWS RNA-binding protein 1 (EWSR1) to facilitate EWSR1 interaction with MYC-associated zinc finger protein (MAZ). The EWSR1-MAZ complex, in turn, transcriptionally upregulates CUX1 expression. Disrupting this *circ-CUX1*/EWSR1/MAZ axis suppresses glycolysis and the growth and metastasis of neuroblastoma cells in vivo [68]. It is noteworthy to mention that high circ-CUX1 and CUX1 expression in neuroblastoma tumors and cell lines is not associated with *MYCN* amplification [68], suggesting that this is an MYCN-independent mechanism for promoting glycolysis in neuroblastoma cells. In another study, it was reported that hepatocyte nuclear factor 4 alpha (HNF4A) and its derived long-noncoding RNA (HNF4A-AS1) promote aerobic glycolysis in neuroblastoma cells by upregulating HK2 and the major glucose transporter SLC2A1 (also known as GLUT1) [69]. Moreover, it was found that high HNF4A-AS1 expression is associated with *MYCN* amplification in neuroblastoma tumors and that MYCN overexpression or knockdown in neuroblastoma cell lines increases or downregulates HNF4A-AS1 expression, respectively [69]. These findings reveal an MYCN-dependent mechanism for promoting glucose import to sustain glycolysis in neuroblastoma cells.

The conversion of pyruvate to lactate is catalyzed by lactate dehydrogenase (LDH) (Figure 1). Human cells express three LDH isoforms, LDHA, LDHB, and LDHC [70,71]. A study by Qing et al. showed that LDHA expression is increased in *MYCN*-amplified neuroblastoma tumors and that MYCN is required for maintaining LDHA expression in neuroblastoma cells [64]. However, a later study by Dorneburg et al. provided evidence that LDHA expression is independent of the *MYCN* amplification status and MYCN expression levels, as determined by a combination of gene expression profiling, immunohistochemistry of neuroblastoma tumors, and MYCN overexpression studies [72]. Nevertheless, both studies showed that LDHA is required for neuroblastoma cell growth and tumorigenicity. Interestingly, the study by Dorneburg et al. reported that knockout of LDHA has no significant effect on glucose consumption and lactate production in neuroblastoma cells in culture and in xenograft tumors. Furthermore, knockdown of LDHB expression in LDHA knockout cells also fails to produce a significant effect on glucose consumption and lactate production. Of note, these neuroblastoma cells do not express significant levels of LDHC. These findings suggest a functional role for LDHA and LDHB in neuroblastoma, independent of aerobic glycolysis [72], but also raise the provocative question of how lactate production is maintained in neuroblastoma cells with no significant levels of LDH expression. Stable isotope tracing experiments using uniformly labeled ^13^C-glucose may help address the source of lactate in these LDH-depleted neuroblastoma cells and underlying mechanisms.

### 3.2. Reprogramming of Amino Acid Metabolism in Neuroblastoma

Of the 20 proteinogenic amino acids, 9 cannot be synthesized by humans and, thus, are considered essential amino acids, including histidine, isoleucine, leucine, lysine, methionine, phenylalanine, threonine, tryptophan, and valine. These essential amino acids can enter cells via amino acid transport [73]. It has been reported that MYCN can transcriptionally activate the expression of the solute carrier family (SLC) of genes, *SLC7A5* and *SLC43A1*, through direct binding to the E-box sequence within both genes. Knockdown of SLC7A5 or SLC43A1 expression in neuroblastoma cells represses the uptake of essential amino acids, as determined by ^3^H-Leucine transport assay, leading to a significant decrease in the intracellular levels of isoleucine, leucine, phenylalanine, and valine. Moreover, SLC7A5 or SLC43A1 depletion reduces MYCN expression by interfering with *MYCN* mRNA translation and attenuates neuroblastoma cell growth in vitro and in vivo. These findings suggest that MYCN and SLC7A5/SLC43A1 form a positive feedback loop to amplify their expression for sustaining the growth and tumorigenicity of neuroblastoma cells [74]. 

However, amino acid metabolic reprogramming in cancer cells is primarily concerned with the eleven non-essential amino acids, including alanine, arginine, asparagine, aspartate, cysteine, glutamate, glutamine, glycine, proline, serine, and tyrosine [75]. Activating transcription factor 4 (ATF4) is a master transcriptional regulator of amino acid synthesis and transport [76,77,78]. In neuroblastoma cells, ATF4 cooperates with the histone lysine demethylase KDM4C to transcriptionally upregulate the expression of enzymes for synthesis of eight non-essential amino acids, including alanine, arginine, asparagine, aspartate, cysteine, glutamate, glycine, and serine. ATF4 and KDM4C also cooperate to induce mRNA expression of *SLC1A4*, *SLC1A5*, *SLC3A2*, *SLC6A9*, *SLC7A1*, *SLC7A5*, and *SLC7A11*, which are involved in the transport of most amino acids [79]. Analysis of patient data reveals significantly higher *ATF4* mRNA levels in neuroblastoma tumors with *MYCN* amplification [80] and higher ATF4 expression is associated with poor prognosis in neuroblastoma patients [81]. Further investigation demonstrates that MYCN binds to the promoter of *ATF4* to increase its mRNA expression [80]. Together, these findings suggest that the MYCN-ATF4 axis, in cooperation with epigenetic regulators, including KDM4C, is a major mechanism to increase the pools of amino acids for sustaining the MYCN-mediated growth program in neuroblastoma [80,82].

Below, we highlight our current understanding of the molecular mechanisms for increasing the uptake and biosynthesis of some key amino acids in neuroblastoma.

#### 3.2.1. Glutamine Metabolism

It has long been established that glutamine is critical for the growth of cancer cells [83,84,85,86]. Highly proliferating cancer cells show predominantly aerobic glycolysis, in which most of the glycolytic pyruvate is converted to lactate instead of feeding into the TCA cycle [58,59,60]. As a result, cancer cells must find other sources of anaplerosis to sustain the TCA cycle for producing ATP and for supplying intermediates for macromolecule biosynthesis. Glutamine, as the most abundant circulating amino acid in the blood and tissues [87], is a major source of anaplerosis [88]. It has a five-carbon backbone and two nitrogen atoms, which serves as a carbon donor to drive the TCA cycle via anaplerosis and a nitrogen donor for biosynthesis of nucleotides and non-essential amino acids [89,90,91] (Figure 2). Increased glutamine metabolism has been considered a “hallmark of cancer metabolism” [89].

Enhanced glutamine metabolism is crucial for the survival and proliferation of neuroblastoma cells. Qing et al. reported that glutamine deprivation triggers apoptosis in *MYCN*-amplified neuroblastoma cell lines by inducing pro-apoptotic proteins PUMA, NOXA, and TRIB3 [92]. Subsequent investigation revealed that *MYCN*-amplified neuroblastoma cells express high levels of ASCT2 (also known as SLC1A5), which functions as a major transporter for uptake of glutamine in *MYCN*-amplified neuroblastoma cells. Increased ASCT2 expression is associated with advanced stages of neuroblastoma and poor prognosis in neuroblastoma patients. The upregulation of ASCT2 is mediated by MYCN and ATF4, which bind the *ASCT2* promoter to induce its transcription. Knockdown of ASCT2 expression markedly inhibited glutamine uptake and decreased the intracellular levels of downstream metabolites of glutaminolysis, including glutamate and the TCA cycle intermediates succinate and fumarate. Moreover, ASCT2 knockdown induced G1 arrest and apoptosis in *MYCN*-amplified neuroblastoma cell lines and significantly inhibited their ability to form a tumor in immunodeficient mice [93]. 

In addition to promoting glutamine uptake, MYCN activates glutaminolysis in neuroblastoma cells [94], which is a process of replenishing the TCA cycle by converting glutamine to α-ketoglutarate [89,90,91]. During glutaminolysis, glutamine is first converted to glutamate by removing its γ-amino group, which is commonly catalyzed by glutaminases (GLS in the cytosol and GLS2 in the mitochondria). It has been shown that MYCN stimulates the conversion of glutamine to glutamate in MYCN-amplified neuroblastoma cells by directly activating *GLS2* transcription [95]. Alternatively, glutamine can be converted to glutamate through amino acid and nucleotide synthesis pathways. Asparagine synthetase (ASNS), a direct transcriptional target of ATF4 [76], converts glutamine and aspartate to glutamate and asparagine. In purine synthesis, two glutamine molecules donate two nitrogen atoms for the synthesis of one molecule of inosine monophosphate (IMP), generating two molecules of glutamate. The synthesis of guanosine monophosphate (GMP) from IMP requires an additional nitrogen atom from glutamine, producing one more molecule of glutamate. Pyrimidine synthesis generates two glutamate molecules: the first step consumes one nitrogen atom from glutamine and the synthesis of CTP from UTP needs one more nitrogen atom from glutamine. All these reactions in purine and pyrimidine nucleotide synthesis are catalyzed by glutamine amidotransferases (PPAT, PFAS, GMPS, CAD, and CTPS1/2), which are transcriptionally activated by MYCN in neuroblastoma cells [94,96]. The functional significance of these enzymatic reactions is underscored by the observation that overexpression of MYCN markedly sensitizes non-*MYCN*-amplified neuroblastoma cells to the glutamine analogue DON (6-diazo-5-oxo-L-norleucine), which blocks the conversion of glutamine to glutamate by inhibiting glutamine amidotransferases [66]. 

Following the conversion of glutamine to glutamate, the next step in utilizing glutamine carbon for cellular bioenergetic and biosynthetic needs is to convert glutamate to α-ketoglutarate to replenish the TCA cycle. Through the TCA cycle, glutamine-derived carbon can be used to drive NADH and FDAH_2_ production and to generate citrate and oxaloacetate. Citrate is a source of acetyl-CoA for the biosynthesis of fatty acids and cholesterol and oxaloacetate is a substrate for the biosynthesis of aspartate [89,90,91]. The conversion of glutamate to α-ketoglutarate is catalyzed by glutamate dehydrogenases (GLUD1 and GLUD2) or transaminases, including glutamate-oxaloacetate transaminase (GOT), glutamate-pyruvate transaminase (GPT), and phosphoserine transaminase 1 (PSAT1). These transaminases use the nitrogen atom from glutamate for the biosynthesis of the non-essential amino acids aspartate (GOT and GOT2), alanine (GPT and GPT2), and serine and glycine (PSAT1). MYCN, ATF4, and KDM4C all contribute to transcriptional upregulation of GOT1, GOT2, GPT, GPT2, and PSAT1 in neuroblastoma cells [80,82,94]. 

In addition to enhancing glutamine transport and glutaminolysis, MYCN may promote glutamine synthesis in neuroblastoma cells. Using MYCN-inducible expression neuroblastoma cell lines and stable isotope tracing with U-^13^C_5_-glucose, Oliynyk et al. showed that MYCN induction increases glutamine synthesis from glucose via α-ketoglutarate [65]. The functional significance of the MYCN-driven glutamine synthesis in neuroblastoma cells remains to be determined. 

Together, these findings suggest that MYCN, ATF4, and KDM4C work together to promote glutamine uptake and metabolism in *MYCN*-amplified neuroblastoma cells to meet the biosynthetic demands of cell growth and proliferation.

#### 3.2.2. Serine and Glycine Metabolism

It was reported more than 60 years ago that serine is important in supporting the growth of human cancer cells in a minimal medium supplemented with dialyzed serum [97]. Cells can obtain serine via transport and de novo biosynthesis. Upregulation of serine and glycine synthesis is an integral part of cancer metabolism [98,99,100,101]. The serine-glycine synthesis pathway consists of four reactions that are catalyzed sequentially by PHGDH, PSAT1, phosphoserine phosphatase (PSPH), and serine hydroxymethyltransferase (SHMT1 in the cytosol and SHMT2 in the mitochondrion). This metabolic pathway links the glycolytic intermediate 3-phosphoglycerate (3PG) to the production of serine, glycine, and the one-carbon carrier 5,10-methylenetetrahydrofolate (5,10-MTHF), along with NADH and α-ketoglutarate (Figure 3). These products are involved in many cellular processes, essential for the growth of cancer cells, including macromolecule synthesis (proteins, nucleic acids, and lipids), redox homeostasis, and methylation regulation. Serine is required for the synthesis of cysteine and sphingolipids; glycine contributes carbon and nitrogen atoms to purine synthesis and is a component of glutathione; and 5,10-MTHF donates carbon to the synthesis of thymidylate and S-adenosylmethionine, the universal methyl donor for methylation reactions. In addition, 5,10-MTHF is a precursor for the synthesis of 10-formyl-THF that donates carbon to purine synthesis [98,99,100,101].

Neuroblastoma tumors with *MYCN* amplification show increased expression of serine and glycine synthesis enzymes, including PHGDH, PSAT1, and SHMT2. The increased expression is primarily the result of transcriptional activation, mediated by both MYCN and ATF4. These two transcription factors can directly bind the promoters of serine-glycine pathway genes to activate their transcription. Moreover, MYCN and ATF4 form a positive feedback loop, with MYCN activating ATF4 mRNA expression and ATF4 stabilizing MYCN protein by suppressing FBXW7-mediated MYCN ubiquitination and degradation [80]. This positive feedback loop presumably reinforces the hyperactive transcription state of these genes to boost production outputs of the serine-glycine synthesis pathway.

ATF4 also cooperates with enzymes that control histone methylation states for transcriptional activation of serine-glycine pathway genes in neuroblastoma cells. G9A (also known as KMT1C) is an H3K9 methyltransferase that has a major role in catalyzing histone H3 lysine 9 monomethylation (H3K9me1) and H3K9me2 in euchromatin [102,103,104,105]. H3K9me1 is an active mark and H3K9me2 a repressive mark [106,107]. It has been reported that G9A is essential for maintaining serine-glycine pathway genes in an active state marked by H3K9me1 [81]. G9A knockdown or inhibition represses the expression of serine-glycine synthesis enzymes, leading to a significant decrease in intracellular serine and glycine levels and autophagic cell death in neuroblastoma cells. This cell death phenotype can be rescued by supplemental serine. Higher G9A expression, which is associated with poor prognosis in neuroblastoma patients, increases serine production from glucose and enhances the proliferation and tumorigenicity of neuroblastoma cells. Importantly, ATF4 is required for G9A to upregulate the serine-glycine pathway. The molecular basis for the ATF4-G9A cooperation remains to be elucidated. These findings uncover an epigenetic program in the control of serine and glycine synthesis in neuroblastoma cells [81].

The histone lysine methylation state is controlled not only by methyltransferases but also by demethylases. Members of the histone lysine demethylase family 4 (KDM4A-D) can remove di- (me2) and tri-methyl (me3) groups from H3K9 [106,107,108]. Thus, activation of the KDM4 family members could increase H3K9me1 levels by removing the repressive markers H3K9me2 and H3K9me3, thereby activating gene expression [108]. Indeed, it was found that KDM4C can epigenetically activate serine-glycine pathway genes under steady-state and serine deprivation conditions by removing the repressive marker H3K9me3. Again, this action of KDM4C requires ATF4. KDM4C activates *ATF4* transcription and, in turn, ATF4 protein interacts with KDM4C to recruit it to the promoters of serine-glycine pathway genes for transcriptional activation [79].

Glycine is the end product in the serine-glycine synthesis pathway. Increased activation of this pathway could lead to the accumulation of glycine, which is detrimental to cells [109]. Glycine can be converted to the toxic metabolites aminoacetone and methylglyoxal [110]. In addition, glycine accumulation can reduce the flux of glucose carbon into the serine-glycine synthesis pathway [110,111]. The glycine cleavage system (GCS) has a major role in glycine breakdown in the mitochondria, which generates CO_2_, NH_3_, NADH, and the one-carbon unit 5,10-MTHF [112,113]. The GCS is composed of glycine decarboxylase (GLDC), aminomethyltransferase (AMT), dihydrolipoamide dehydrogenase (DLD), and glycine cleavage system protein H (GCSH). The first and rate-limiting step in GCS-mediated glycine breakdown is catalyzed by GLDC. The importance of the GCS in the maintenance of glycine homeostasis is evidenced by genetic findings that in mice and humans, *GLDC* mutations cause glycine encephalopathy (also known as non-ketotic hyperglycinemia) and neural tube defects, as a result of glycine accumulation [114,115,116].

Alptekin et al. reported that *GLDC* and serine-glycine pathway genes are co-upregulated in *MYCN*-amplified neuroblastoma tumors and cell lines. It was further shown that *GLDC* is a direct transcriptional target gene of MYCN. Depletion of GLDC by RNA interference disrupts multiple metabolic processes, affecting the TCA cycle and production of amino acids, purine nucleotides, and lipids. As a result, GLDC knockdown markedly inhibits the proliferation and tumorigenicity of *MYCN*-amplified neuroblastoma cell lines. These findings suggest a key role for glycine clearance in driving central carbon metabolism and biosynthesis of nucleotides and lipids in neuroblastoma cells [111].

In summary, the presented studies suggest that MYCN and ATF4 coordinate with the histone H3 methyltransferase G9A and the demethylase KDM4C to transcriptionally activate the serine-glycine biosynthesis pathway for increasing the production of serine, glycine, and one-carbon units in neuroblastoma cells.

#### 3.2.3. Cysteine Metabolism

Cysteine is a sulfur-containing amino acid. Aside from being a proteinogenic amino acid, cysteine and its catabolic products are involved in many important cellular processes. Cysteine provides a highly reactive thiol side chain that functions as a nucleophile critical for the catalytic activity of many enzymes. The formation of disulfide bonds between cysteines is crucial for protein folding and stability. Further, cysteine is one of the three amino acid components of glutathione that counteracts reactive oxygen species (ROS) [117]. Moreover, cysteine catabolism produces hydrogen sulfide (H_2_S) to feed the electron transport chain for ATP production and pyruvate to supply carbon to the TCA cycle [118].

Cysteine can be synthesized de novo via the transsulfuration pathway, in which the enzyme cystathionine b-synthase (CBS) catalyzes the condensation of the methionine cycle intermediate homocysteine with serine to form cystathionine, which is then hydrolyzed by cystathionine γ-lyase (CTH) to generate cysteine and α-ketoglutarate. However, the major source of intracellular cysteine is the import of extracellular cystine coupled to the efflux of intracellular glutamate via the system Xc^−^ transporter [119], which is composed of xCT (SLC7A11) that confers cystine transport function and 4F2hc (also known as SLC3A2) that serves as a chaperone to recruit SLC7A11 to the plasma membrane [73,120]. Once in the cytosol, cystine is readily reduced to produce two molecules of cysteine. 

The expression of CBS, CTH, SLC3A2, and SLC7A11 is transcriptionally upregulated by KDM4C and ATF4 in neuroblastoma cells, leading to a significant increase in the intracellular levels of cysteine [79]. A recent study also reported that MYCN transcriptionally upregulates the system Xc^−^ transporter to meet the increased demand of *MYCN*-amplified neuroblastoma cells to detoxify ROS by glutathione. This dependency creates a vulnerability that can be targeted by ferroptosis inducers [121]. Along the same lines, a more recent study showed that MYCN has a key role in maintaining cellular cysteine pools and that high MYCN expression sensitizes neuroblastoma cells to cystine deprivation, leading to lipid peroxidation and ferroptosis [122]. Together, these recent studies uncover an MYCN-dependent vulnerability to ferroptosis induction.

### 3.3. Reprogramming of Nucleotide Metabolism in Neuroblastoma

Intracellular nucleotide pools are maintained by de novo synthesis and salvage. Quiescent cells primarily rely on salvage pathways for supplying degraded bases for nucleotide synthesis [123,124]. However, proliferating cells must increase de novo production of nucleotides to meet the demands of DNA replication, ATP generation, and RNA (mRNA, rRNA, and tRNA) expression for protein biosynthesis. This metabolic reprogramming involves multiple metabolic pathways that provide building blocks for nucleotide biosynthesis, including the pentose phosphate pathway, amino acid and one-carbon metabolism, the TCA cycle, and electron transport and OXPHOS [124,125].

#### 3.3.1. Purine Metabolism

In mammalian cells, de novo purine nucleotide synthesis is catalyzed by six enzymes (Figure 4), including phosphoribosyl pyrophosphate amidotransferase (PPAT), the trifunctional enzyme glycinamide ribonucleotide synthetase-aminoimidazole ribonucleotide synthetase-glycinamide ribonucleotide transformylase (GART), phosphoribosyl formylglycinamidine synthase (PFAS), and the bifunctional enzymes phosphoribosylaminoimidazole carboxylase (PAICS) and AICAR transformylase-IMP cyclohydrolase (ATIC) [124,125]. The starting substrate is 5-phosphoribosyl-1-pyrophosphate (PRPP), which is synthesized by PRPP synthase (PRPS1 and PRPS2) using ATP and ribose 5-phosphate generated by the pentose phosphate pathway. PPAT catalyzes the first and rate-limiting reaction that converts PRPP to 5-phosphoribosylamine (PRA). The end product of de novo purine synthesis is IMP, which is the precursor for all purine nucleotides. The purine base in IMP is composed of four nitrogen atoms (two from glutamine, one from aspartate, and one from glycine) and five carbons (two from glycine, two from the one-carbon carrier 10-formyl-THF, and one from CO_2_). The salvage pathway uses free purine nucleobases in the form of adenine, guanine, and hypoxanthine. These free bases are attached to PRPP by adenine phosphoribosyltransferase (APRT) to generate AMP or by hypoxanthine-guanine phosphoribosyltransferase (HPRT), which acts on hypoxanthine to form IMP and on guanine to form GMP [126].

MYCN has a major role in promoting purine biosynthesis in neuroblastoma cells. MYCN increases the production of glycine and 10-formyl-THF by transcriptionally upregulating enzymes in the serine-glycine-one-carbon network, with contributions from ATF4, KDM4C, and G9A [79,80,81]. MYCN also increases mRNA expression of enzymes involved in de novo purine synthesis, including PRPS1, PPAT, PFAS, PAICS, ADSL, and IMPDH1 [96]. In addition, chromatin immunoprecipitation and sequencing using anti-MYCN antibody provided evidence that MYCN directly targets *PAICS* and *MTHFD2* for transcriptional upregulation [127]. MTHFD2 catalyzes the conversion of 5,10-MTHF to 10-formy-THF for donating carbon to purine ring synthesis (Figure 3). 

#### 3.3.2. Pyrimidine Metabolism

The primary function of pyrimidine nucleotides is to serve as building blocks for RNA and DNA synthesis. In addition, pyrimidine nucleotides have a key role in carbohydrate and lipid metabolism. Uridine diphosphate (UDP) sugars are substrates for all glycosylation reactions and glycogen synthesis and cytidine diphosphate (CDP)-diacylglycerol is required for the biosynthesis of complex glycerolipids [128,129,130]. In humans, pyrimidine nucleotides are produced by a combination of de novo biosynthesis and salvage (Figure 4). The trifunctional enzyme, composed of carbamoyl phosphate synthase, aspartate transcarbamoylase, and dihydroorotase (CAD), catalyzes the first step in de novo biosynthesis that generates dihydroorotate from glutamine, aspartate, and bicarbonate. Dihydroorotate is then oxidized to orotate by dihydroorotate dehydrogenase (DHODH), a mitochondrial membrane protein. This reaction is coupled to a reduction in ubiquinone (coenzyme Q, CoQ) to ubiquinol (CoQH_2_), linking pyrimidine nucleotide production to mitochondrial electron transport [131]. Next, the bifunctional enzyme UMP synthetase (UMPS) catalyzes the reaction that converts orotate into uridine monophosphate (UMP), which, in turn, serves as the precursor to generate all other pyrimidine nucleotides for RNA and DNA synthesis and carbohydrate and lipid metabolism. The salvage pathway recycles UMP and cytidine monophosphate (CMP) derived from intracellular RNA degradation or imports nucleosides (uridine and cytidine) from the bloodstream. Uridine-cytidine kinase (UCK) then converts uridine and cytidine into UMP and CMP, respectively. The pyrimidine ring is composed of four carbons (three from aspartate and one from bicarbonate) and two nitrogen atoms (one from glutamine and one from aspartate) [128,129].

MYCN coordinates multiple metabolic pathways to promote pyrimidine nucleotide synthesis. It activates de novo synthesis by transcriptional upregulation of CAD, DHODH, and UMPS [96]. It also cooperates with ATF4 to increase substrates for de novo synthesis by enhancing glutamine transport via transcriptional upregulation of SLC1A5 [93] and by increasing aspartate synthesis via upregulation of GOT1 and GOT2 [80,94]. In addition, MYCN increases UCK2 expression to activate the salvage pathway [96]. Moreover, MYCN transcriptionally upregulates the enzymes for dTMP synthesis, including thymidylate synthase and SHMT2 for producing 5,10-MTHF [80,96]. 

Collectively, the reported findings reveal an essential role for MYCN in enhancing nucleotide synthesis by transcriptional activation of synthesis enzymes and other metabolic pathways that provide substrates for nucleotide production. As a result, MYCN overexpression significantly increases intracellular pools of nucleotides in neuroblastoma cells [96].

### 3.4. Reprogramming of Lipid Metabolism in Neuroblastoma

#### 3.4.1. Fatty Acid Metabolism

Increased fatty acid metabolism promotes cancer development and progression through a variety of mechanisms, including membrane biosynthesis, energy storage and production, and signaling transduction. Reprogramming of lipid metabolism in cancer cells includes changes in fatty acid transport, de novo fatty acid synthesis, storage as lipid droplets, and fatty acid β-oxidation to generate acetyl-CoA for driving the TCA cycle and ATP production [132,133] (Figure 5).

Cells can import fatty acids, primarily in the form of long-chain fatty acids, through the family of fatty acid transport proteins (FATP1-6), encoded by *SLC27A1-6* [134,135]. A recent study provided evidence for MYCN in the regulation of fatty acid transport [136]. A combination of untargeted metabolomics and targeted lipidomics reveals glycerolipid accumulation in *MYCN*-amplified neuroblastoma cell lines and primary tumors. Further investigation identified *SLC27A2*, encoding FATP2, as a direct target gene of MYCN for transcriptional upregulation in neuroblastoma cells, leading to increased fatty acid uptake for glycerolipid synthesis. SLC27A2-mediated fatty acid uptake is crucial for the survival of *MYCN*-amplified neuroblastoma cells. Genomic knockout of *SLC27A* or pharmacological inhibition of SLC27A2 significantly attenuates the growth of *MYCN*-amplified neuroblastoma cell lines and tumors. Moreover, SLC27A2 inhibition synergizes with chemotherapy in limiting neuroblastoma growth in various animal models. These findings uncover an MYCN-induced dependency on fatty acid transport in neuroblastoma with therapeutic implications [136].

De novo fatty acid synthesis uses cytoplasmic acetyl-CoA as a substrate (Figure 5). Glycolysis, fatty acid β-oxidation, acetate conversion, and glutamine are significant sources of acetyl-CoA in cancer cells [137,138,139,140]. In the mitochondria, acetyl-CoA is condensed with oxaloacetate to form citrate, which can be exported to the cytosol and cleaved by ATP citrate lyase (ACLY) to produce cytosolic acetyl-CoA. Acetyl-CoA carboxylases (ACACs) then catalyze carboxylation of acetyl-CoA to form malonyl-CoA, which is followed by fatty acid synthase (FASN)-catalyzed successive condensation of seven malonyl-CoA molecules and one priming acetyl-CoA to generate the 16-carbon saturated fatty acid palmitate. Further elongation and desaturation of palmitate, carried out, respectively, by elongases and desaturases, produce fatty acids with varying lengths and degrees of saturation that serve as substrates for the synthesis of fatty-acid-containing lipids, including cellular membranes and signaling molecules.

Fatty acid synthesis is controlled by sterol regulatory element-binding proteins (SREBPs), which are also bHLH-Zip transcription factors, including SREBP1a, SREBP1c, and SREBP2. SREBP1a and SREBP1c are produced from a single gene, *SREBF1*, through alternative transcription start sites and SREBP2 is encoded by *SREBF2*. SREBP1c and SREBP2 are primarily responsible for transcriptional upregulation of enzymes involved in fatty acid and cholesterol synthesis, respectively, whereas SREBP1a is a transcriptional activator of all SREBP-responsive genes [141]. 

A study by Carroll et al. provides evidence for MYCN in promoting fatty acid synthesis [142]. MYCN transcriptionally upregulates the expression of MondoA (also known as MLXIP) in neuroblastoma cells [142], which is a member of the extended MYC network with a role in regulation of cellular metabolism [143]. MondoA is required for maintaining the expression of SREBP1. Knockdown of MYCN or MondoA markedly inhibits de novo fatty acid synthesis. Importantly, blocking fatty acid synthesis through the FASN inhibitor C75 induces cell death only in high-MYCN-expressing cells [142], indicating a crucial role for fatty acid synthesis in sustaining the survival of *MYCN*-amplified neuroblastoma cells.

Fatty acids are degraded via β-oxidation (FAO) (Figure 5) [133,144,145,146], which is a significant source of energy, generating twice as much ATP as carbohydrates based on their dry mass. The first step in FAO converts fatty acids to acyl-CoAs (activated fatty acids), which is catalyzed by members of the acyl-CoA synthetase long-chain family (ACSL). Next, acyl-CoAs are converted to acyl-carnitines by carnitine palmitoyl transferase 1 (CPT1). This is the rate-limiting step in FAO and is required for transporting fatty acids across the outer-mitochondrial membrane. Once in the mitochondria, acyl-carnitines are reverted to acyl-CoAs, which then undergo a cyclical series of reactions that result in progressive shortening of fatty acids by two carbons per cycle, generating NADH, FADH_2_, and acetyl-CoA. The last cycle produces two acetyl-CoA molecules from a four-carbon fatty acid. FAO-derived NADH and FADH_2_ enter the electron transport chain to produce ATP and acetyl-CoA enters the TCA cycle to produce ATP and building blocks for macromolecule biosynthesis. 

MYCN has a key role in promoting FAO in *MYCN*-amplified neuroblastoma cells. Blocking MYCN activity by the small-molecule inhibitor 10058-F4 leads to lipid accumulation in the form of lipid droplets. Proteomic analysis revealed that 10058-F4 treatment downregulates many enzymes involved in FAO, including ACADM, DECR1, ECHS1, ECI1, HADHA, HADHB, and HSD17B10. Inhibition of FAO could recapitulate the lipid accumulation phenotype [147]. Moreover, analysis of neuroblastoma patient data revealed that high expression of CPT1C is associated with poor prognosis [65], although it is not clear whether MYCN has a role in its upregulation. Inhibition of FAO by small-molecule inhibitors that target CPT1C reduces the survival of *MYCN*-amplified cells in vitro and attenuates tumor growth in vivo [65]. These findings suggest that MYCN-enhanced FAO supports neuroblastoma growth, which could be targeted for therapy.

#### 3.4.2. Cholesterol Metabolism

The mevalonate pathway produces sterols (e.g., cholesterol) and non-sterol isoprenoids [148,149] (Figure 5). Cholesterol biosynthesis is orchestrated by more than 20 enzymes. The first step is catalyzed by the cytoplasmatic enzyme acetoacetyl-CoA thiolase 2 (ACAT2) that joins two acetyl-CoA molecules to form acetoacetyl-CoA. This is followed by a reaction catalyzed by 3-hydroxy-3-methylglutaryl-CoA synthase (HMGCS), which introduces the third molecule of acetyl-CoA to form the branched-chain molecule 3-hydroxy-3-methylglutaryl-CoA (HMG-CoA). HMG-CoA is then reduced to mevalonate in the first rate-limiting step in cholesterol biosynthesis that is catalyzed by HMG-CoA reductase (HMGCR). Mevalonate is further phosphorylated to isopentyl pyrophosphate (IPP), which is converted to geranyl pyrophosphate. Condensation with another IPP yields farnesyl pyrophosphate (FPP). Farnesyl-diphosphate farnesyltransferase 1 (FDFT1, also known as squalene synthase) is the first specific enzyme in cholesterol biosynthesis, catalyzing the dimerization of two molecules of FPP to yield squalene. Squalene epoxidase (SQLE), the other rate-limiting enzyme in cholesterol biosynthesis, converts squalene into its epoxydic form 2,3-epoxysqualene, which is then cyclized to form lanosterol by lanosterol synthase (LSS). Finally, lanosterol is converted to cholesterol in a series of enzymatic reactions. In non-sterol reactions, FPP is converted to geranylgeranyl-PP (GGPP) and both FPP and GGPP are used for prenylation and activation of the RAS superfamily of GTPases [150]. Isoprenoids are also used to produce dolichol, heme A, and ubiquinone (CoQ), which have important functions in many cellular processes [151,152]. 

SREBP2, encoded by *SREBF2*, is primarily responsible for transcriptional activation of mevalonate pathway enzymes, low-density lipoprotein (LDL) receptor, and mediators of cholesterol efflux. SREBP2 is synthesized as an inactive precursor bound to the endoplasmic reticulum (ER) membrane in a complex with SREBP-cleavage activating protein (SCAP). When sterol is abundant, the SCAP-SREBP2 complex binds tightly to the ER resident insulin-induced gene 1 or 2 protein (INSIG1/2), leading to SREBP2 retention in the ER. When the sterol level is low, the SCAP-SREBP2 complex dissociates from INSIGs and translocates to the Golgi, where SREBP2 is cleaved, releasing its N-terminal transcription factor domain, which then enters the nucleus and activates the transcription of mevalonate pathway genes. This end-product feedback control is crucial for keeping the mevalonate pathway in check to maintain sterol homeostasis [153,154,155].

Cholesterol is the precursor to produce all steroid hormones, bile acids, and vitamin D. It is also an essential structural component in the cellular membrane, modulating fluidity and creating membrane rafts for concentrating signaling molecules [156]. Thus, continuous production of cholesterol is required for cancer cell proliferation. Increased mevalonate pathway products are a common feature of cancer metabolism [149,157,158]. The importance of the mevalonate pathway in cancer development is underscored by the recent finding that blocking this pathway is a major mechanism for the tumor suppressor function of p53 [159], which is encoded by *TP53*, the most frequently mutated gene in human cancer [160]. Targeting the mevalonate pathway and cholesterol metabolism has been increasingly recognized as a promising therapeutic strategy against cancer [161].

Increased activation of the mevalonate pathway is also a metabolic feature of high-risk neuroblastoma. In a study to investigate the cellular basis of high-risk neuroblastoma, Liu et al. isolated a minor population of cells from primary neuroblastoma tumors developed in *TH-MYCN* mice. These cells grow as spheres in the culture medium for primary neural crest cells and possess self-renewal, differentiation, and high tumorigenic potential. Gene expression profiling of independent lines of sphere-forming cells in comparison with their parental primary tumors revealed metabolic reprogramming in sphere-forming cells that is characterized by transcriptional upregulation of all genes within the cholesterol and serine-glycine biosynthetic pathways. This upregulation is associated with the stem-cell or proliferative state, as expression of these pathway genes is markedly downregulated following retinoic-acid-induced differentiation. Importantly, blocking the mevalonate pathway by statins, a class of cholesterol-lowering drugs that inhibit Hmgcr (Figure 5), induces growth arrest and cell death in sphere-forming cells. The increased activation of the mevalonate pathway is mediated by Srebp proteins. Both Srebp1 and Srebp2 are upregulated in sphere-forming cells and blocking of Srebp activation abrogates the transcriptional activation of the mevalonate pathway [82]. 

The observations from the study of mouse neuroblastoma sphere-forming cells are relevant to human neuroblastoma. Transcriptional activation of the mevalonate pathway is also observed in human high-risk neuroblastoma in comparison with low-risk tumors, which is independent of the *MYCN* amplification status. Both *MYCN*-amplified and non-*MYCN* neuroblastoma cell lines are sensitive to statins [82], suggesting that statins are potential therapeutics for high-risk neuroblastoma.

## 4. Conclusions

In this review, we discuss recent advances in our understanding of metabolic reprogramming in neuroblastoma cells. Despite the progress, there are major areas that remain largely unexplored. We know very little about the metabolic reprogramming in high-risk neuroblastoma tumors that carry no *MYCN* amplification. Thus, it is unclear to what extent metabolic reprogramming varies between *MYCN*-amplified and non-*MYCN* high-risk tumors. There are indications that they may be different. For example, compared to low-risk tumors, non-*MYCN* high-risk tumors show no significant upregulation of the serine-glycine synthesis pathway, whereas in *MYCN*-amplified tumors and cell lines, this pathway is highly active, as evidenced by higher expression of the pathway enzymes and increased flux of glucose carbon through the pathway [80]. In contrast to serine-glycine synthesis, increased activation of the mevalonate pathway is observed in high-risk neuroblastoma tumors, independent of the *MYCN* amplification status [82]. A molecular understanding of the metabolic differences between *MYCN*-amplified and non-*MYCN* high-risk tumors will shed light on their pathogenesis and suggest new therapeutic strategies.

In addition to *MYCN*-amplification states, neuroblastoma tumors display varying degrees of differentiation and have different sets of somatically acquired mutations [4,7]. Thus, they must have heterogenous metabolic dependencies that need to be identified and targeted specifically. Knowledge of the metabolic flexibility of neuroblastoma cells, i.e., the capacity to rewire their metabolism in response to inactivation of individual metabolic pathways or processes, is also crucial to the development of effective metabolism-based therapies. Finally, to be successful in targeting neuroblastoma metabolism for therapy, we need to gain a better understanding of how neuroblastoma metabolism is influenced by the tumor microenvironment that is composed of distinct cell populations and microbiome.

Recent studies on super enhancers in neuroblastoma tumors and cell lines may help address these issues. Super enhancers are clusters of enhancers with high enrichment for the binding of master transcription factors, coactivators, and epigenetic regulators, which promote high-level gene expression [162,163]. Profiling of super enhancers and their target genes has led to molecular identification of super-enhancer-driven neuroblastoma subtypes that match known clinical groups, including *MYCN*-amplified, non-*MYCN*-amplified high-risk, and low-risk neuroblastoma tumors [164]. Similar investigations with neuroblastoma cell lines also reveal two or more neuroblastoma cell types with distinct gene expression profiles [163,165]. These studies provide a molecular basis for the cellular heterogeneity in neuroblastoma tumors and a rich source of molecular information for identifying subtype-associated metabolic reprogramming.

Nevertheless, even the limited progress so far in the study of neuroblastoma metabolism has already revealed new therapeutic opportunities (Table 1). For example, MYCN-mediated metabolic reprogramming generates dependencies that can be targeted by blocking glutamine transport [93], utilization [166] and anaplerosis [94,167], by inhibition of serine-glycine synthesis [80], by blocking pyrimidine transport and production [96], and by depletion of cysteine [122]. As our understanding of neuroblastoma metabolism grows, this will reveal new metabolic hubs for therapeutic interventions. It has become increasingly clear that there is a coupled relationship between metabolic reprogramming and metabolic vulnerabilities: cancer cells reprogram their metabolism to meet the biosynthetic challenge of growth but, as a result, are inherently sensitive to metabolic perturbation. This coupled relationship may offer numerous opportunities in targeting metabolism for cancer therapy.

## Figures and Tables

**Figure 1 cancers-14-04113-f001:**
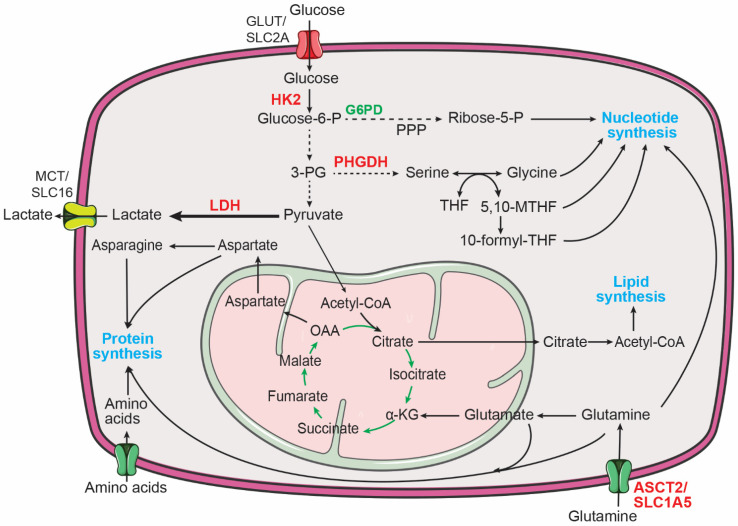
Major metabolic pathways for supporting macromolecule biosynthesis in cancer cells. Increased central carbon metabolism provides building blocks for the production of proteins, nucleotides, and lipids to sustain the proliferation of cancer cells. 3-PG, 3-phosphoglycerate; 5,10-MTHF, 5,10-methylene-tetrahydrofolate; 10-formyl-THF, 10-formyl-tetrahydrofolate; α-KG, α-ketoglutarate; Glucose-6-P, glucose-6-phosphate; G6PD, glucose-6-phosphate dehydrogenase; HK2, hexokinase 2; LDH, lactate dehydrogenase; OAA, oxaloacetate; PHGDH, phosphoglycerate dehydrogenase; PPP, pentose phosphate pathway; THF, tetrahydrofolate. MYCN targets are highlighted in red.

**Figure 2 cancers-14-04113-f002:**
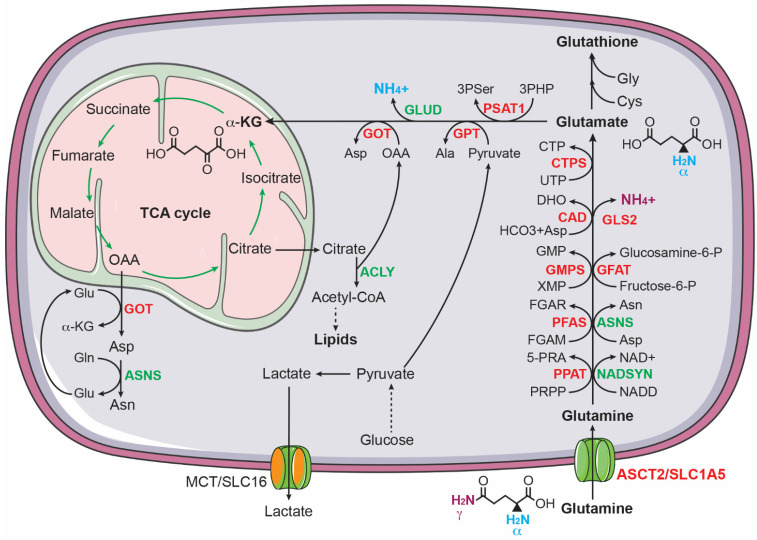
Glutamine uptake and catabolism supports biosynthesis in cancer cells. Glutamine donates its γ-nitrogen for the biosynthesis of asparagine (Asn), nucleotides, NAD^+^, and glucosamine-6-phosphate (glucosamine-6-P) and its α-nitrogen for 3-phospho-serine (3Pser), alanine (Ala), and aspartate (Asp). Glutamine also provides carbon via α-ketoglutarate to sustain the TCA cycle for producing ATP and intermediates for the biosynthesis of aspartate, asparagine, and lipids. ASNS, asparagine synthetase; CAD, carbamoyl phosphate synthase, aspartate transcarbamoylase, and dihydroorotase; CTPS, CTP synthase; GFAT, glutamine fructose-6-phosphate amidotransferase; GMPS, guanine monophosphate synthase; GOT, glutamic-oxaloacetic transaminase; GPT, glutamic--pyruvic transaminase; NADSYN, NAD synthetase; PFAS, phosphoribosyl formylglycinamidine synthase; PPAT, phosphoribosyl pyrophosphate amidotransferase. MYCN targets are highlighted in red.

**Figure 3 cancers-14-04113-f003:**
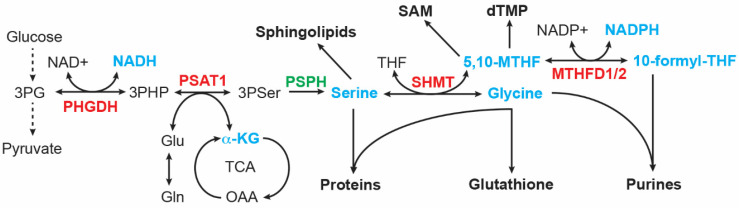
Serine-glycine synthesis pathway. Intermediates and products (highlighted in cyan) of this metabolic pathway serve as building blocks for the synthesis of proteins, lipids, nucleotides, reducing equivalents (NADH and NADPH), glutathione, and S-adenosyl-methionine (SAM). MTHFD, Methylenetetrahydrofolate dehydrogenase; PHGDH, phosphoglycerate dehydrogenase; PSAT1, phosphoserine Aminotransferase 1; PSPH, phosphoserine phosphatase; SHMT, serine hydroxymethyltransferase. MYCN targets are highlighted in red.

**Figure 4 cancers-14-04113-f004:**
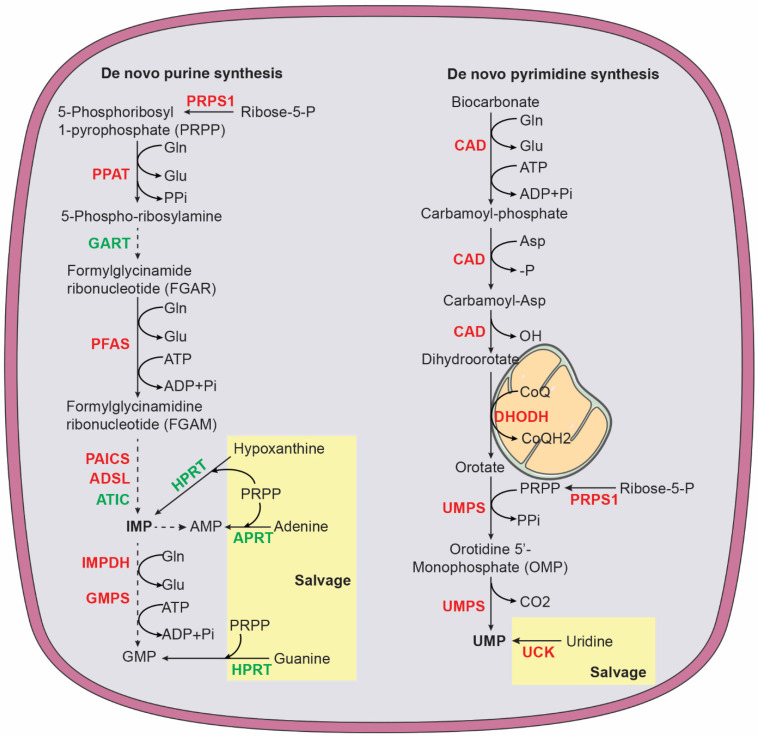
Purine and pyrimidine biosynthesis via de novo and salvage pathways. Purine pathway: ADSL, adenylosuccinate lyase; APRT, adenine phosphoribosyltransferase; ATIC, 5-aminoimidazole-4-carboxamide ribonucleotide formyltransferase and IMP cyclohydrolase; GART, phosphoribosylglycinamide formyltransferase, phosphoribosylglycinamide synthetase, phosphoribosylaminoimidazole synthetase; Glu, glutamate; Gln, glutamine; HPRT, hypoxanthine-guanine phosphoribosyltransferase; IMPDH, inosine monophosphate dehydrogenase; PRPS1, phosphoribosyl pyrophosphate synthetase 1; PAICS, phosphoribosyl aminoimidazole carboxylase and phosphoribosyl aminoimidazole succinocarboxamide synthase; PFAS, phosphoribosyl formylglycinamidine synthase; PPAT, phosphoribosyl pyrophosphate amidotransferase. Pyrimidine pathway: CAD, carbamoyl phosphate synthase, aspartate transcarbamoylase, and dihydroorotase; CoQ ubiquinone; CoQH_2_, ubiquinol; DHODH, dihydroorotate dehydrogenase; UCK, uridine-cytidine kinase; UMPS, UMP synthetase. Enzymes highlighted in red are MYCN targets.

**Figure 5 cancers-14-04113-f005:**
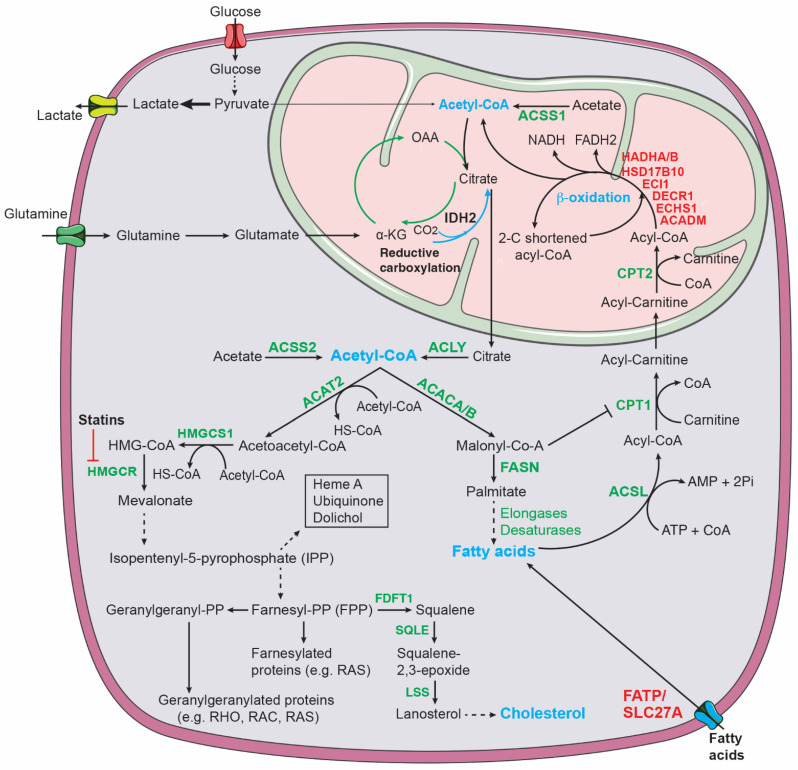
Lipid metabolic pathways. Glucose, acetate, and fatty acid β-oxidation contribute to the mitochondrial pool of acetyl-CoA that can be exported to the cytosol via citrate for lipid synthesis. ACSS1 (acyl-CoA synthetase short chain family member 1) and ACSS2 convert acetate to acetyl-CoA in the mitochondria and cytosol, respectively. Glutamine-derived α-KG can generate citrate via either oxidation in which α-KG-derived OAA condenses with acetyl-CoA to form citrate or reductive carboxylation in which α-KG is reduced to citrate. IDH2 (isocitrate dehydrogenase 2) catalyzes reductive carboxylation in the mitochondria. Cholesterol synthesis: ACAT2, acetyl-CoA acetyltransferase 2; FDFT1, farnesyl-diphosphate farnesyltransferase 1; HMG-CoA, 3-hydroxy-3-methylglutaryl CoA; HMGCR, HMG-CoA reductase; HMGCS1, HMG-CoA synthase 1; LSS, lanosterol synthase; SQLE, squalene epoxidase. Fatty acid synthesis and β-oxidation: ACACA/B, acetyl-CoA carboxylase alpha/beta; ACSL, acyl-CoA synthetase long chain family; ACADM, acyl-CoA dehydrogenase medium chain; CPT1/2; carnitine palmitoyltransferase 1/2; DECR1, 2,4-dienoyl-CoA reductase 1; ECHS1, enoyl-CoA hydratase, short chain 1; ECI1, enoyl-CoA delta isomerase 1; FASN, fatty acid synthase; FATP, fatty acid transport protein; HADHA/B, hydroxyacyl-CoA dehydrogenase trifunctional multienzyme complex subunit alpha/beta; HSD17B10, hydroxysteroid 17-beta dehydrogenase 10. MYCN targets are highlighted in red.

**Table 1 cancers-14-04113-t001:** Examples of small molecules targeting MYCN-driven metabolism reprogramming.

Metabolic Pathway	MYCN Target	Inhibitor
Glycolysis	HK2 [64]	2-Deoxy-D-Glucose (2-DG) [168,169] Benitrobenrazide [170]
	LDHA [64]	GNE-140 [171] FX11 [172,173] Galloflavin [174]
Glutamine metabolism	GLS2 [95]	Compound 968 [175]
Serine-Glycine synthesis	PHGDH [80]	NCT503 [176] CBR-5884 [177] PKUMDL-WQ-2101 [178] PH-755 [179]
	SHMT2 [80]	SHIN2 [180] AGF347 [181]
Nucleotide synthesis	Glutamine amidotransferases (PPAT, PFAS, GMPS, CAD, and CTPS1/2) [96]	6-diazo-5-oxo-L-norleucine (DON) [182]
	PPAT [96]	6-Mercaptopurine [183]
	IMPDH [96]	VX-944 [184] Tiazofurin [184] FF-10501 [185]
	DHODH [96]	Leflunomide [186,187] Brequinar [188,189] GSK983 [190]
Lipid metabolism	FATP2 (SLC27A2) [136]	Lipofermata [191,192] Grassofermata (CB5) [136,191]

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
