# Peer review of "MYCN and Metabolic Reprogramming in Neuroblastoma"

_cancers, 2022, doi:10.3390/cancers14174113_

Round 1

Reviewer 1 Report

In this manuscript, Banal et al. review how MYCN reprogram cell metabolism in a pediatric cancer, neuroblastoma. The authors systemically and clearly introduce major metabolic pathways including glycolysis, amino acid, nucleotide and lipid metabolism. The authors further present convincing evidence showing that MYCN transcriptionally upregulates various key metabolic enzymes in hallmark of cancer metabolism by cooperation with transcription factor ATF4 and epigenetic regulators including histone lysine demethylase KDM4C and histone H3 methyltransferase G9A. However, there are a few points that need to be more precisely addressed.

1.       On page 4, 2nd paragraph, the authors describe a novel mechanism (circ-CUX1/EWSR1/MAZ axis) plays a role in upregulating aerobic glycolysis. Is this mechanism MYCN-dependent or independent?

2.       On page 10, Figure 4, there are some inconsistencies between labels in the figure and description in the text. For example, GART and ATIC in the text not shown in the figure; HPRT (in the figure) not consistent with HGRT in the text (the first paragraph in 3.3.1).

3.       On page 15, 3rd paragraph, the authors addressed that the tumor suppressor p53 has a novel function by blocking the mevalonate pathway to affect cancer development. Does p53 still sustain this novel function in neuroblastoma with MYCN amplification?

4.       Fatty acid oxidation (FAO) is upregulated in neuroblastoma, blocking MYCN activity/inhibiting FAO reduces the survival of MYCN-amplified cells.  Does MYCN directly affect transcription of key enzymes in FAO pathway to reprogram FAO?

5.       Finally, the authors have overlooked the role of MYC proteins as transcriptional amplifiers – thus they can increase overall transcription (causing hypertranscription).  Has any work been done on transcriptional amplification/hypertranscription in neuroblastoma cells and its dependency on MYCN? Hypertranscription is part of the key role MYC proteins play in driving macromolecular synthesis.  It may not need a separate section in the review, but should at least be mentioned as another important entry point into metabolic control by MYCN.   

Author Response

Reviewer 1

In this manuscript, Banal et al. review how MYCN reprogram cell metabolism in a pediatric cancer, neuroblastoma. The authors systemically and clearly introduce major metabolic pathways including glycolysis, amino acid, nucleotide and lipid metabolism. The authors further present convincing evidence showing that MYCN transcriptionally upregulates various key metabolic enzymes in hallmark of cancer metabolism by cooperation with transcription factor ATF4 and epigenetic regulators including histone lysine demethylase KDM4C and histone H3 methyltransferase G9A. However, there are a few points that need to be more precisely addressed.

Response:  We thank the reviewer for careful reading of our manuscript and positive comments and constructive suggestions. Please see below for our point-by-point responses.

  1. On page 4, 2nd paragraph, the authors describe a novel mechanism (circ-CUX1/EWSR1/MAZ axis) plays a role in upregulating aerobic glycolysis. Is this mechanism MYCN-dependent or independent?

Response: We appreciate the reviewer’s comment. As indicated in the paper, high circ-CUX1 and CUX1 expression in neuroblastoma tumors and cell lines is not as-sociated with MYCN amplification. Accordingly, we have revised text to incorporate this point: “It is noteworthy to mention that high circ-CUX1 and CUX1 expression in neuroblastoma tumors and cell lines is not associated with MYCN amplification, suggesting that this is a MYCN-independent mechanism for promoting glycolysis in neuroblastoma cells” (page 4, last paragraph).

For the report on HNF4A and HNF4A-AS1, evidence was presented for MYCN dependency. Accordingly, we have revised text to clarify the point: “Moreover, it was found that high HNF4A-AS1 expression is associated with MYCN amplification in neuroblastoma tumors and that MYCN overexpression or knockdown in neuroblastoma cell lines increases or downregulates HNF4A-AS1 expression, respectively. These findings reveal a MYCN-dependent mechanism for promoting glucose import to sustain glycolysis in neuroblastoma cells” (page 4, last paragraph).

  1. On page 10, Figure 4, there are some inconsistencies between labels in the figure and description in the text. For example, GART and ATIC in the text not shown in the figure; HPRT (in the figure) not consistent with HGRT in the text (the first paragraph in 3.3.1).

            Response: We thank the reviewer for careful reading of our manuscript. We have corrected all these errors.

  1. On page 15, 3rd paragraph, the authors addressed that the tumor suppressor p53 has a novel function by blocking the mevalonate pathway to affect cancer development. Does p53 still sustain this novel function in neuroblastoma with MYCN amplification?

            Response: We have conducted literature search and found no refences on p53 regulation of cholesterol metabolism in neuroblastoma with MYCN amplification.

  1. Fatty acid oxidation (FAO) is upregulated in neuroblastoma, blocking MYCN activity/inhibiting FAO reduces the survival of MYCN-amplified cells. Does MYCN directly affect transcription of key enzymes in FAO pathway to reprogram FAO?

            Response: We thank the reviewer for raising this issue. Blocking MYCN activity by the small molecule inhibitor 10058-F4 resulted in downregulation of several enzymes in the FAO pathway. We have revised text and Figure 5 to include these enzymes. In addition, we have revised text to point out that there is no evidence that MYCN regulates CPT1C expression in neuroblastoma (page 16, 2nd paragraph).

  1. Finally, the authors have overlooked the role of MYC proteins as transcriptional amplifiers – thus they can increase overall transcription (causing hypertranscription). Has any work been done on transcriptional amplification/hypertranscription in neuroblastoma cells and its dependency on MYCN? Hypertranscription is part of the key role MYC proteins play in driving macromolecular synthesis.  It may not need a separate section in the review, but should at least be mentioned as another important entry point into metabolic control by MYCN.  

            Response: We thank the reviewer for raising this important issue. We found one paper (Cell, 2014) by the Rani George lab that directly demonstrates transcriptional amplification by MYCN in neuroblastoma cell lines. We have revised text to incorporate this point:

“In addition to the classic model of MYCN-mediated transcriptional regulation, recent evidence indicates that MYCN amplification or high-level expression can drive global transcriptional amplification to increase the production of transcripts from all active genes in neuroblastoma cells, which is mediated by MYCN invading of the promoters and enhancers of active genes [57]. Since many enzyme-encoding genes are actively transcribed for the maintenance of basic metabolic functions, it is reasonable to speculate that the MYCN-driven global transcriptional amplification could have a profound impact on metabolic reprogramming in neuroblastoma” (page 4, 2nd paragraph).

Reviewer 2 Report

This paper is about the metabolic changes specially caused by MYCN amplification in neuroblastoma. 

This is a highly redundant paper about the metabolic changes in cancer cells and in MYCN amplified neuroblastoma cells. The paper gives an almost  full-coverage of metabolism and changes in normal and tumor cells, quite hard to internalize the full information.

It is not always clear what changes are related especially to NMYC amplification only as the title of the paper requests and what changes occur in all (neuroblastoma) cancer cells. Unfortunately, Figures are general description of metabolic pathway without any special signs in terms of  alteration in MYCN amplified neuroblastoma.

The last sentence of the Abstract "We also briefly discuss some important areas that need to be explored for successful development of metabolism-based therapy against high-risk neuroblastoma." is not really covered by the manuscript.

I would suggest also a Table, where the special changes of metabolism caused by MYCN amplification in neuroblastoma are revealed point by point with possible therapeutic approach.

Detailed comments:

In neuroblastoma serum LDH is usually highly elevated, in 3.1 section  paragraph 3 authors describes that controversy information about LDH(ABC) expression, which is not really high, this question should be at minimum theoretically solved.

3.2 Paragraph 1. "SLC7A5 and SLC43A1 depletion reduces  MYCN expression", some lines later, "crucial role of MYCN-mediated upregulation of SLC7A5 and SLC43A1..." which regulate which, it seems contradictory.

3.42. Paragraph 4: "mevalonate pathway  is also a metabolic feature of high-risk neuroblastoma" . It is a strange sentence in this paper as MYCN amplified neuroblastoma is not equivalent with high risk neuroblastoma

Author Response

Reviewer 2

We thank the reviewer for his/her comments and suggestions. Please see below for our point-by-point responses.

This paper is about the metabolic changes specially caused by MYCN amplification in neuroblastoma.

This is a highly redundant paper about the metabolic changes in cancer cells and in MYCN amplified neuroblastoma cells. The paper gives an almost full-coverage of metabolism and changes in normal and tumor cells, quite hard to internalize the full information.

            Response:  The goal of this manuscript is to give a comprehensive review of metabolic reprogramming in neuroblastoma. We are not aware of recent reviews covering this topic as comprehensive as ours. Any references on other cancers are solely for the purpose of background information.

It is not always clear what changes are related especially to NMYC amplification only as the title of the paper requests and what changes occur in all (neuroblastoma) cancer cells. Unfortunately, Figures are general description of metabolic pathway without any special signs in terms of alteration in MYCN amplified neuroblastoma.

            Response: We thank the reviewer for raising this point and have revised all figures in which MYCN targets are highlighted in red.

As mentioned above, this review is about metabolic reprogramming in neuroblastoma, regardless of MYCN-dependent and -independent mechanisms. “MYCN” in the title is to emphasize the major role of MYCN in neuroblastoma metabolic reprogramming. It is also because we know very little about metabolic reprogramming in non-MYCN neuroblastoma. Finally, we hope that our review is of interest to readers who may not be interested in neuroblastoma but MYC family members in metabolic reprogramming.

            The “general description of metabolic pathways” in our figures is to help readers who have limited background on metabolism and to make it easier for them to follow the content of this review.

The last sentence of the Abstract "We also briefly discuss some important areas that need to be explored for successful development of metabolism-based therapy against high-risk neuroblastoma." is not really covered by the manuscript.

            Response: We see the point raised by the reviewer. We have replaced “briefly discuss” with “highlight”.

To summarize: In the last section “Conclusion”, we list three major areas that remain largely unexplored: 1) metabolic reprogramming in high-risk neuroblastoma tumors that carry no MYCN amplification, 2) metabolic heterogeneity and flexibility of neuroblastoma cells, and 3) impact of tumor microenvironment on neuroblastoma metabolic reprogramming. In this revised manuscript, we have added a paragraph briefly describing recent studies on super enhancer-driven neuroblastoma cell and tumor types, which may help address these issues.

I would suggest also a Table, where the special changes of metabolism caused by MYCN amplification in neuroblastoma are revealed point by point with possible therapeutic approach.

            Response: We thank the reviewer for the suggestion. As mentioned above, we have revised all figures to highlight MYCN target genes. In addition, we have added a table showing examples of MYCN targets involved in metabolism and their small molecule inhibitors for experimental and therapeutic investigations.

Detailed comments:

In neuroblastoma serum LDH is usually highly elevated, in 3.1 section paragraph 3 authors describes that controversy information about LDH(ABC) expression, which is not really high, this question should be at minimum theoretically solved.

            Response: Yes, it has been reported that sera from neuroblastoma patients show high levels of LDH. But it is not clear to us how this observation may explain the findings reported by the cited study in which the combination of LDHA knockout and LDHB knockdown (and no LDHC expression) in neuroblastoma cells has no significant effect on aerobic glycolysis, lactase activity, and lactate production. The reported LDH assay was conducted with cellular extracts, which largely eliminated the possibility of contamination with serum LDH (also the cell culture used normal FBS). We simply suggest a metabolic assay that, from our point of view, may help address this question.

            Also, we do not consider the findings controversial but rather “provocative” (our word) or “intriguingly” (authors’ word).

3.2 Paragraph 1. "SLC7A5 and SLC43A1 depletion reduces MYCN expression", some lines later, "crucial role of MYCN-mediated upregulation of SLC7A5 and SLC43A1..." which regulate which, it seems contradictory.

            Response: We thank the reviewer for raising this issue. The cited study reports that MYCN transcriptionally upregulates the expression of SLC7A5 and SLC43A1, and SLC7A5 and SLC43A1 are required for MYCN mRNA translation by importing essential amino acids. We have revised text to clarify the issue:

“These findings suggest that MYCN and SLC7A5/SLC43A1 form a positive feedback loop to amplify their expression for sustaining the growth and tumorigenicity of neuroblastoma cells”.

3.42. Paragraph 4: "mevalonate pathway is also a metabolic feature of high-risk neuroblastoma". It is a strange sentence in this paper as MYCN amplified neuroblastoma is not equivalent with high risk neuroblastoma.

            Response: As mentioned above, our manuscript is about metabolic reprogramming in neuroblastoma, regardless of whether MYCN is involved or not.